# Associations between Personality Traits, Intolerance of Uncertainty, Coping Strategies, and Stress in Italian Frontline and Non-Frontline HCWs during the COVID-19 Pandemic—A Multi-Group Path-Analysis

**DOI:** 10.3390/healthcare9081086

**Published:** 2021-08-23

**Authors:** Ramona Bongelli, Carla Canestrari, Alessandra Fermani, Morena Muzi, Ilaria Riccioni, Alessia Bertolazzi, Roberto Burro

**Affiliations:** 1Department of Political Science, Communication and International Relations, University of Macerata, 62100 Macerata, Italy; alessia.bertolazzi@unimc.it; 2Department of Education, Cultural Heritage and Tourism, University of Macerata, 62100 Macerata, Italy; carla.canestrari@unimc.it (C.C.); alessandra.fermani@unimc.it (A.F.); morena.muzi@unimc.it (M.M.); ilaria.riccioni@unimc.it (I.R.); 3Department of Human Sciences, University of Verona, 37129 Verona, Italy; roberto.burro@univr.it

**Keywords:** COVID-19, HCWs, personality traits, intolerance of uncertainty, coping strategies, perceived stress

## Abstract

The COVID-19 pandemic represented a very difficult physical and psychological challenge for the general population and even more for healthcare workers (HCWs). The main aim of the present study is to test whether there were significant differences between frontline and non-frontline Italian HCWs concerning (a) personality traits, intolerance of uncertainty, coping strategies and perceived stress, and (b) the models of their associations. A total of 682 Italian HCWs completed a self-report questionnaire: 280 employed in COVID-19 wards and 402 in other wards. The analysis of variance omnibus test revealed significant differences between the two groups only for perceived stress, which was higher among the frontline. The multi-group path analysis revealed significant differences in the structure of the associations between the two groups of HCWs, specifically concerning the relations between: personality traits and intolerance of uncertainty; intolerance of uncertainty and coping strategies. Regarding the relation between coping strategies and stress no difference was identified between the two groups. In both of them, emotionally focused coping was negatively related with perceived stress, whereas dysfunctional coping was positively related with stress. These results could be useful in planning actions aiming to reduce stress and improve the effectiveness of HCWs’ interventions. Training programs aimed to provide HCWs with a skillset to tackle uncertain and stressful circumstances could represent an appropriate support to develop a preventive approach during outbreaks.

## 1. Introduction

The COVID-19 pandemic is undoubtedly one of the greatest disasters of the 21st century for people all over the world. The pandemic is a very serious threat, both physical and psychological, for the general population [1,2,3,4,5], but even more for vulnerable groups of subjects. The latter include, among others, patients with pre-existent mental health disorders, as well as those suffering from other chronic or acute diseases, such as cancer patients. While COVID-19 and the related strict lockdown caused, since the first wave of the pandemic, severe psychological effects (such as relapses, worsening of conditions, stress, anger, impulsivity, etc., e.g., [6,7,8,9]) on patients suffering from mental disorders, it has been particularly challenging also for cancer patients, who are at a high risk of contracting the virus and of developing more severe complications compared to the general population [10]. Thus, the fear of being infected makes the COVID-19 a new stressor [11], able to affect their emotional and social functioning [12].

At the same time, the COVID-19 pandemic represents a very arduous challenge for both the scientific community—involved in finding vaccines to prevent its spread, and therapies to cure infected people [13]—and healthcare workers (HCWs) —who, working daily in facing it, not only jeopardize their own physical health (risking to get infected), but also their mental wellbeing. Consistently with the results of many studies on the psychological impact of past pandemics on health professionals [14,15,16], the literature published until now about COVID-19 revealed that HCWs are at particular risk of adverse psychological outcomes, i.e., of developing more severe mental symptoms, including stress, anxiety, depression, distress, insomnia, emotional exhaustion, burnout, as well as post-traumatic stress disorder [17,18,19,20,21,22,23,24,25,26,27,28,29,30,31,32,33,34,35,36,37,38,39,40]. These adverse consequences regard specifically those working on the frontline [17] (i.e., those directly engaged in the diagnosis, treatment, and care for patients with COVID-19, employed in emergency departments, intensive care units, and infectious disease wards) and in areas (such as China and Italy) where the virus has had a rapid spread and caused a high number of hospitalizations (in intensive care units) and deaths (especially during the first months of its circulation).

These negative psychological outcomes on HCWSs are undoubtedly related to the situation—that, specifically during the first wave of the virus spread, was in itself particularly demanding [24], stressful, and characterized by high levels of uncertainty—but probably also to more specific contextual conditions (first of all, as argued above, having worked on the frontline or not), as well as to individual differences, such as HCWs’ personality traits (that are “one of the important determinants for the development of mental health issues during the pandemic situation” [41] (p. 5)), ability to tolerate uncertainty, and to cope with the situation. 

### 1.1. Intolerance of Uncertainty, Personality, and Coping

All pandemics, including the one caused by COVID-19, being unexpected and unpredictable events, which affect large numbers of people, are sources of stress (i.e., they are cataclysmic stressors, according to Lazarus and Cohen’s definition [42]), and uncertainty among both ordinary people and HCWs [43,44,45]. If ordinary people experience “uncertainty about getting infected, uncertainty about the seriousness of the infection, uncertainty about whether the people around you are infected, uncertainty about whether objects or surfaces (e.g., money, doorknobs) are infected, uncertainty about the optimal type of treatment or protective measures, and uncertainty about whether a pandemic is truly over” [44] (p. 43), HCWs experience also other types of uncertainty (both professional and personal), that differ during the different stages of virus diffusion. Among them, uncertainty about how dangerous and contagious the virus is, uncertainty about therapies and cures, uncertainty concerning personal devices to be adopted in order to avoid getting infected while working (and become a vehicle of infection for other people, e.g. patients or relatives), uncertainties about the right measures for containing the virus spread (e.g., use of masks and gloves), etc.

Although uncertainty during a pandemic is, therefore, a common experience for everybody, including HCWs, nonetheless the individuals’ abilities to tolerate it varies greatly. Some people, more than others, show indeed more difficulty in tolerating uncertainty. 

Intolerance of uncertainty can be defined as a dispositional fear of the unknown [46,47,48], which seems to be related to certain personality traits [49], specifically to neuroticism (or negative emotionality, which is one of the five personality traits identified by the BIG Five Model [50]; see Section 2.2. “Measures”). It can be considered as a sub-trait of anxiety [44] (that is, in its turn, a facet of neuroticism), which has often been found in association with stress, distress, insomnia, psychosomatic symptoms, and other clinical conditions in several recent studies carried out on COVID-19 among the general population and HCWs (e.g., [45,51,52,53,54,55,56]). It is a cognitive, emotional, and behavioral tendency to react negatively to uncertain or ambiguous situations and unpredictable future events [57,58], which biases information processing, leading to faulty appraisals of threat, and reduces coping abilities [59]. 

### 1.2. Coping, Personality, Intolerance of Uncertainty

Coping, in its turn, can be defined as the set of cognitive and behavioral efforts to manage specific external and/or internal demands, which are evaluated as taxing or exceeding a person’s resources [60] or, more simply, as processes of response to stressors [61]. Like Monzani et al. [62] state, coping strategies have been classified differently, mainly in dichotomous pairs, by different authors: problem-focused, (i.e., aiming at actively responding to a stressful situation) vs. emotion-focused coping (i.e., aiming to reduce or manage emotions related to the stressful situation) [63,64]; approach, (i.e., aiming to directly face with stressors and related emotions) vs. avoidance strategies (i.e., aiming to deny, minimize, or avoid dealing with stressors) [65,66,67]; adaptive (i.e., characterized by more probability of obtaining a result) vs. ineffective or maladaptive (i.e., characterized by more probability of not obtaining a result) [67,68]). 

The vast literature on this topic revealed that the use of coping strategies is influenced by many variables, among which are situational demands, environmental and cultural aspects, personal characters [63], as well as individuals’ ability to tolerate uncertainty [59], and personality traits [69,70,71]. In other terms, different persons, in different situations, resort to different coping strategies. 

Many recent studies during the COVID-19 pandemic have been conducted in different contexts among both the general population and health professionals, showing great variability in the use of coping strategies. Taylor et al. [72], for example, revealed that during the lockdown, people have found many different ways of making self-isolation more tolerable, which include watching TV or movies. Regarding HCWs, Munawar and Choundry [73], for example, identified different types of coping strategies used by Malaysian HCWs to deal with stress and anxiety, but one of the most recurring was the religion coping strategy. Salman et al. [74] found, in a sample of HCWs from Pakistan, that positive coping strategies were more widely used than avoidant and maladaptive strategies. Huang et al. [75], comparing nurses with nursing students, found that the former use more problem-focused coping strategies than the latter. 

As mentioned above, the use of different coping strategies is not only linked to specific contextual, environmental, or cultural conditions, it also seems to be influenced by individuals’ dispositional traits. As for the link between coping strategies and intolerance of uncertainty, although much research has been conducted revealing clear associations between them [59], as far as we know, few studies focused on the relations between intolerance of uncertainty and coping strategies during a pandemic and none of them explicitly analyzed this relation in samples of HCWs. One of the best-known research about intolerance of uncertainty and coping was the one conducted by Taha et al. [76] in a general population sample during the H1N1 pandemic of 2009. The authors found significant relations between, greater intolerance of uncertainty, on the one hand, and lower problem-focused and higher emotion-focused coping strategies, on the other. Instead, Rettie and Daniels [77], studying a sample of the general population during the COVID-19 pandemic, found that maladaptive coping strategies mediate the relationship between intolerance of uncertainty and distress.

As far as coping strategies and personality are concerned (see Section 2.2 for the personality traits), scientific research not only revealed that personality influences the way people cope with stressful situations, but identified also specific relations between them. For example, according to Leandro and Castillo [70], and Afshar et al. [71], maladaptive personality traits (e.g., neuroticism) positively correlate with emotion-focused and avoidant (dysfunctional) coping strategies; on the contrary, extraversion positively correlates with problem-focused and emotional-focused strategies [69]. Several recent studies on COVID-19 have also identified similar links between personality traits and adaptive and maladaptive coping responses. Sica et al. [78], for example, found, in a sample of Italian adults, positive association between maladaptive traits of personality and avoidant forms of coping (e.g., drug use), and negative associations between maladaptive traits and acceptance and positive reframing. Other studies have not only substantially confirmed these results, but have also found significant associations with the levels of perceived stress. According to Liu et al. [79], for example, individuals with higher levels of neuroticism would have the tendency “to perceive events as highly threatening and often have limited coping resources, self-regulation and perceived efficacy, and thus resulting in a higher level of stress” [79] (p. 2). Conversely, people with high levels of conscientiousness seem to be able to resort to more effective coping strategies, thus experiencing lower levels of stress. 

### 1.3. Current Study

Consistently with the results of the literature on the topic, it seems reasonable to assume that specific contextual situations as well as some individual characteristics—i.e., personality traits, intolerance of uncertainty, coping strategies—have specific relations among them and differently impact on psychological outcomes, specifically on perceived stress. Nonetheless, to the best of our knowledge, no study has been conducted to explicitly investigate these associations in samples of HCWs during the COVID-19 pandemic. 

Thus, also considering the great amount of work, under uncertain and stressful conditions, the present study aimed at investigating, in a sample of Italian HCWs, employed during the first wave of the COVID-19 pandemic, the relations between some personal characteristics and perceived stress, by testing whether the variable “having worked on the frontline” (i.e., in a COVID dedicated ward) or “not having worked on the frontline” (i.e., in other wards) affects them (i.e., affects the relations between the variables taken into account).

The analyses revealed significant differences both in the levels of perceived stress, which were higher in the frontline HCWs than in the non-frontline, and in the structure of the associations between the two groups, specifically concerning the relations between: personality traits and intolerance of uncertainty; intolerance of uncertainty and coping strategies. Regarding the relations between coping strategies and stress, no difference was identified. In both groups, the use of emotional coping strategies was linked indeed to lower levels of perceived stress, while the use of dysfunctional coping strategies to higher levels of perceived stress.

## 2. Methods

### 2.1. Data Collection

This study, conducted according to Helsinki Declaration principles (https://www.wma.net/what-we-do/medical-ethics/declaration-of-helsinki/, accessed on 22 August 2021), APA Ethics Code, and European and Italian Privacy Law (i.e., EU Reg. 679/2016, GDPRD and Legislative Decree n. 196/2003, Code regarding the protection of personal data), has been approved by PhD meeting curriculum in Psychology, Communication, and Social Sciences, (University of Macerata. Protocol code n. 19435, 3 August 2020).

It was conducted through an online survey, which started on May 15 and ended on 30 July 2020, to which Italian HCWs (nurses and physicians), enrolled in professional orders and associations, were invited to participate. The snowball sampling method was used. 

Specifically, during the first week of May 2020, the authors sent an email to the presidents of all the Italian orders of physicians and nurses and of the main professional associations (e.g., Associazione Anestesisti Rianimatori Ospedalieri Italiani emergenza area critica/Italian Hospital Anesthetist Association for critical area emergency) to present the research protocol and asked them to send an invitation email to their members with the link to compile a self-report questionnaire or to publish it on their website. After three weeks, the authors sent a reminder email to those orders and associations that did not respond to the first email.

It should be noted that before sending the emails and making public the link to the online survey, three physicians, one obstetrician, and one nurse compiled the questionnaire and provided the authors their favorable opinion regarding its length, clarity, and comprehensibility of all the items. Survey administration was conducted through LimeSurvey software on a LAMP (Linux, Apache, MySQL, PHP, a common example of a web service stack) server. All communication was encrypted, using HTTPS protocol and Secure Sockets Layer (SSL). No prize, as an incentive to compile the survey was offered, since it is not common practice in Italy and it could also affect the data by inducing subjects to offer socially desirable answers, thus compromising data reliability.

The questionnaire opened with some information concerning the aims of the research, the identity and contacts of the research team, the planned ways of disseminating the results, the references to the European and Italian privacy laws, and protection of personal data. The respondents could begin to fill in the questionnaire after having voluntarily consented to participate by signing an online informed consent. The questionnaire was composed of: Twelve questions, aiming to collect socio-demographic, employment information, and information concerning the exposure of HCWs to COVID-19;Four validated scales (see Section 2.2. “Measures”), aiming to measure HCWs’ personality traits, intolerance of uncertainty, coping strategies, and perceived stress;One final open-ended question (which is not taken into account in the present study, as it is the specific subject of another paper that we are going to submit), aiming to know whether and how the experience of having worked during the pandemic had an emotional impact on HCWs.

All the items of the questionnaire were compulsory, except for the open-ended question. The estimated average time for compiling the questionnaire was approximately 15 min.

### 2.2. Measures

#### 2.2.1. Personal Information Data

In order to collect socio-demographic and job characteristics, as well as more specific information concerning their exposure to COVID-19, HCWs were asked questions concerning gender, age, marital and parental status, religion, job position (doctor or nurse), specialties (area), place of work, seniority, job exposure to COVID-19, COVID-19 swabs (i.e., having received swabs for COVID-19), and contraction of COVID-19.

#### 2.2.2. Big Five Inventory, Short Version (BFI-2-S)

The Italian HCWs’ personality trait domains were assessed using the Italian translation of the 30-item BFI-2-S [80]; it is a short version of the 60-item BFI-2 [81], which, in its turn, represents a revision of the Big Five Inventory (BFI, [82,83,84]).

The BFI-2 “operationalizes the hierarchical conceptualization of personality structure by assessing the Big Five domains and 15 facets: Extraversion (with facets of Sociability, Assertiveness, and Energy Level), Agreeableness (Compassion, Respectfulness, and Trust), Conscientiousness (Organization, Productiveness, and Responsibility), Negative Emotionality (Anxiety, Depression, and Emotional Volatility), and Open-Mindedness (Intellectual Curiosity, Aesthetic Sensitivity, and Creative Imagination)” [81] (p. 69). Respondents rate each of the 30 items using a five-point scale ranging from 1 (strongly disagree) to 5 (strongly agree).

The CFA outcomes supported the hypothesized structure: all standardized factor loadings resulted statistically significant (with values between 0.420 and 0.914), and the goodness of fit indexes acceptable (CFI = 0.911; TLI = 0.902; RMSEA = 0.079; SRMR = 0.088). We considered (throughout the article) as fit indexes the comparative fit index (CFI), the TLI, the RMSEA, and the SRMR, with CFI and TLI ≥ 0.90, RMSEA ≤ 0.08, and SRMR ≤ 0.06 as threshold values [85].

Cronbach’s alpha and McDonald’s omega were respectively 0.74 and 0.73 for Extraversion, 0.71 and 0.70 for Agreeableness, 0.70 and 0.69 for Conscientiousness, 0.77 and 0.77 for Negative Emotionality, 0.77 and 0.78 for Open-Mindedness. 

#### 2.2.3. Intolerance of Uncertainty Scale (IUS-12)

The Italian HCWs’ intolerance of uncertainty was assessed using the Italian version [86] of IUS-12 [87]; it is the short version of the 27-item intolerance of uncertainty scale (IUS-27 [88]), developed to evaluate “emotional, cognitive and behavioral reactions to ambiguous situations, implications of being uncertain, and attempts to control the future” [88] (p. 791). IUS-12 is a two-factor scale that represents two different sub-dimensions of intolerance toward uncertainty: prospective and inhibitory [49,87]. The former reflects “desire for predictability and active engagement in information seeking to increase certainty”; the latter reflects “uncertainty avoidance and paralysis in the face of uncertainty” [89] (p. 377). Respondents assess the items on a five-point Likert scale, ranging from 1 (not at all characteristic of me) to 5 (entirely characteristic of me).

Both the Cronbach’s alpha and McDonald’s omega were 0.90 for the overall scale; Cronbach’s alpha for prospective intolerance of uncertainty (items 1–7) was 0.83 and McDonald’s omega was 0.84, while alpha and omega for inhibitory intolerance (items 8–12) were 0.90. 

#### 2.2.4. Brief-COPE Scale

The Italian HCWs’ coping strategies were evaluated using the Brief-COPE [66]; it is the short version of the original COPE (Coping Orientation to Problems Experienced) inventory [65]. We adapted the original Brief-COPE scale into the Italian language using a forward and backward translation process to guarantee correspondence between Italian and English original versions. The Brief-COPE consists of 14 faced-scales (each of them composed of 2 items), which represent 14 different coping strategies [66,67] that can be grouped into two overarching coping styles: approach coping (active coping, planning, positive reframing, acceptance, seeking emotional support, seeking instrumental support) and avoidant coping (self-distraction, denial, venting, substance use, behavioral disengagement, self-blame). Humor and religion are excluded from these styles, since, according to [90], they are both adaptive and problematic components. Some authors (e.g., [63]) distinguish the 14 faced-scales into three composite subscales: problem-focused (active coping, seeking instrumental support, planning), emotion-focused (acceptance, seeking emotional support, humor, positive reframing, religion), and dysfunctional (behavioral disengagement, denial, self-blame, self-distraction, substance use, venting). The 28 items, that are measured with scores ranging from 0 (I haven’t been doing this at all) to 3 (I’ve been doing this a lot), can be “converted to a dispositional ‘coping style’ format […] or a situational concurrent format, by changing verb forms […]. They can assume a retrospective, situational format […], a concurrent situational format […], or even a dispositional format” [66] (pp. 95–98). Since we wanted to measure the Italian healthcare professionals’ situational and retrospective coping strategies, i.e., related to a specific circumstance (the COVID-19 pandemic), we presented the items in the past tense. 

In order to assess the goodness of fit indexes of the factor structure of the Italian version of the Brief-COPE scale, we performed a Confirmatory Factor Analysis (CFA). The CFA outcomes supported the hypothesized structure: all standardized factor loadings resulted statistically significant (with values between 0.430 and 0.989), and the goodness of fit indexes acceptable (CFI = 0.927; TLI = 0.919; RMSEA = 0.078; SRMR = 0.089).

Cronbach alpha for the Brief-COPE was 0.82, and McDonald’s omega was 0.82. Specifically, following the distinction between problem-focused, emotion-focused, and dysfunctional strategies, we obtained that alpha and omega for problem-focused strategies were 0.76 and 0.77, respectively, while for emotion-focused strategies 0.71 and 0.77, respectively. Alpha and omega for dysfunctional strategies were 0.77 and 0.79, respectively.

#### 2.2.5. Italian Perceived Stress Scale (IPSS-10)

The Italian HCWs’ perceived stress was measured using the IPSS-10 (Italian Perceived Stress Scale); it is the Italian version [91] of the PSS (Perceived stress scale). Although its original version consists of 14 items [92], the most commonly used is that consisting of 10 items [93,94]. PSS measures the degree to which situations in one’s life are evaluated as stressful [95] (p. 1) by asking about feelings and thoughts during the last month. Respondents are asked how often they felt a certain way on a five-point Likert scale: 0 = Never; 1 = Almost Never; 2 = Sometimes; 3 = Fairly Often; 4 = Very Often. Cronbach’s alpha was 0.881 and McDonald’s omega 0.884. 

### 2.3. Procedures

In order to investigate the associations between personality traits, intolerance of uncertainty, coping strategies and perceived stress in a sample of Italian HCWs, also taking into account different situational contexts (i.e., having worked on the frontline or not), we

First, tested if there were significant differences between the two groups of HCWs (frontline and non-frontline) in relation to each of the variables considered;Second, developed and tested a model (see Figure 1), according to which personality traits can differentially impact on intolerance of uncertainty, intolerance of uncertainty can differently impact on the use of coping strategies, and coping strategies can differently affect the level of perceived stress;Finally, tested whether the structure of the relations (see Figure 1) vary in the two groups of HCWs. 

### 2.4. Data Analysis

Descriptive statistics (n, %) were conducted, using the R-software (version 4.1.0, [96]), to have a complete picture of our sample. 

In order to determine whether there were significant differences between frontline and non-frontline HCWs, three different linear mixed models (LMMs) were applied using personality traits (based on BFI-2-S score), intolerance of uncertainty (based on IUS-12 score), and coping strategies (emotional focused, problem focused, avoidant; based on Brief-COPE score) as fixed effects, and subject ID as random effect. A linear model (LM) was performed for perceived stress (based on IPSS-10). Three analyses of variance fixed effects omnibus tests (regarding LMMs), and one analysis of variance omnibus test (regarding LM) were calculated.

A multi-group path-analysis was performed, using lavaan R software package [97], in order to test whether having worked in a dedicated COVID ward or not during the first wave of the pandemic in Italy would have determined differences in the associations (i.e., in the structure of relations) between personality traits, intolerance of uncertainty, coping strategies, and perceived stress.

## 3. Results

### 3.1. Sample Characteristics

Out of 682 participants who fully compiled the questionnaire, 530 (77.71%) were women and 152 (22.29%) men. The participants’ answers contained no missing data (answers to individual questions were mandatory to complete the questionnaire). Incomplete questionnaires were not taken into account. The mean age was 45.39 (ranging from 21 to 81, SD = 12.04). The majority of them were married (44.36%), had children (57.92%), declared to be religious persons (practitioners: 15.84%; non-practitioners: 23.75%; only occasional practitioners: 38.42%). Furthermore, 75.95% worked as a nurse, 70.23% in hospitals and care services in the northern region of Italy, 51.76% in the area of medical specialties, with 47.65% working for more than 20 years. Moreover, 41.06% of them claimed to have worked in a COVID-19-dedicated ward, i.e., on the frontline, while 58.94% in other wards. Although more than half of them (57.48%) had a swab test for COVID-19, fortunately, only a low percentage contracted the virus (8.36%). The following Table 1 shows a more detailed description of our sample characteristics. 

### 3.2. Analysis of Variance

The analysis of variance fixed effects omnibus test (Type III analysis of variance with Satterthwaite’s method), conducted on the LMMs, revealed no significant differences between frontline and non-frontline HCWs concerning personality traits, F(4, 2720) = 1.664, *p* = 0.155 (see Figure 2A), intolerance of uncertainty, F(1, 680) = 0.131, *p* = 0.718 (see Figure 2B) and coping strategies, F(1, 1360) = 2.253, *p* = 0.106 (see Figure 3A). 

In particular, in both groups of HCWs

The conscientiousness (i.e., organization, productiveness, and responsibility) was the most prevalent personality trait;Levels of prospective intolerance of uncertainty were higher than the levels of inhibitory one;Emotion-focused coping strategies were more used than problem-focused and dysfunctional coping strategies.

On the contrary, the results of the analysis of variance omnibus test (Type III analysis of variance) conducted on the linear model (LM) revealed significant differences between the two groups about the perceived stress, F(1, 680) = 9.394, *p* = 0.002, Cohen’s d = 0.240, i.e., the quantitative size of the estimated effect is closer to the small than to the medium value. Specifically, the levels of perceived stress were higher among the frontline Italian HCWs (M = 22.032, SD = 8.649, 95% CI [21.447, 22.617]) rather than among the non-frontline (M = 20.042, SD = 8.119, 95% CI [19.584, 20.501]) (see Figure 3B).

In Figure 4, we report the correlations and the descriptive statistics for personality traits, intolerance of uncertainly, coping strategies, and perceived stress. 

### 3.3. Multi-Group Path-Analysis

We ran the multi-group path-analysis (see path-diagram in Figure 1) using the “Diagonally Weighted Least Squares” (DWLS) estimator. For a proper analysis, the minimum ratio between the number of observations and the number of model parameters should be greater than 5:1 [85]. In our case, we had 110 estimated parameters with 682 participants, therefore the ratio was 6.2:1, and the sample size was adequate. We obtained adequate fit indices: CFI = 0.959, TLI = 0.917, RMSEA = 0.077, and SRMR = 0.060. 

Using RMSEA as effect size and alpha = 0.05, the results of the post-hoc power analysis show that a sample size of N = 682 is associated with a power larger than 99.99%.

The Chi-squared difference test between the multi-group unconstrained and constrained models revealed significant differences (see Table 2) between Italian frontline and non-frontline HCWs concerning the associations (structure of the relations) among personality traits, intolerance of uncertainty, coping strategies, and perceived stress.

The arrows in the following Figure 5 show respectively the significant association identified among the Italian frontline (see Figure 5A) and non-frontline HCWs (see Figure 5B). Asterisks indicate the level of significance of the estimated effects. Just by looking at the two figures, it is possible to notice how the structure of the relations among the variables differs noticeably in the two groups of HCWs and is more complex in the non-frontline one.

The analysis revealed significant differences in the structure of the associations between the two groups of HCWs, concerning specifically the relations between:(a)Personality traits and Intolerance of uncertainty. While neuroticism was positively related to inhibitory and prospective intolerance of uncertainty (i.e., the more neuroticism the more intolerance of uncertainty both prospective and inhibitory) in both groups, other significant relations were found exclusively in the non-frontline HCWs. Specifically: conscientiousness was negatively related to prospective intolerance of uncertainty (i.e., the more conscientiousness, the less prospective intolerance of uncertainty. In other words, the more organized, productive, and responsible HCWs are, the less they are engaged in information-seeking to increase certainty), while agreeableness and open mindedness were negatively related to the inhibitory intolerance of uncertainty (i.e., the more agreeableness and open mindedness, the less inhibitory intolerance of uncertainty. In other words, the more confident, and intellectually creative and curious HCWs are, the less they seem to be paralyzed by uncertainty). These results seem to suggest that personality traits of frontline HCWs have a poor influence on levels of intolerance to uncertainty, except for the negative emotionality, which seems to act analogously in both HCWs’ groups.(b)Intolerance of uncertainty and Coping strategies. No significant relation was found in the frontline group of HCWs. Vice versa, in the non-frontline one, while prospective intolerance of uncertainty was positively related to problem and emotion focused coping strategies (i.e., the more prospective intolerance of uncertainty, the more problem and emotion focused coping strategies), inhibitory intolerance of uncertainty was negatively related both to problem and emotion-focused coping (i.e., the more inhibitory intolerance of uncertainty, the less problem and emotion focused coping strategies), and positively related to dysfunctional coping ones (i.e., the more inhibitory intolerance of uncertainty, the more dysfunctional coping strategies).

Regarding the relation between coping strategies and stress no difference was identified between the two groups. In both of them, emotionally focused coping was negatively related with perceived stress (i.e., the more emotion focused strategies, the less stress), whereas the dysfunctional one was positively correlated with stress (i.e., the more dysfunctional strategies, the more stress).

## 4. Discussion and Conclusions

COVID-19 was (and is) an arduous challenge for HCWs all over the world, but especially for those working in the areas characterized by a rapid spread of the virus, which caused, specifically during the first waves, high numbers of hospitalizations (in intensive care units) and deaths, such as in Italy [22]. 

Many research studies, focused on the impacts of COVID-19 on mental health [98], have revealed that HCWs, involved in fronting this pandemic as those engaged during the past ones [14,15,16], were at particular risk of developing severe mental symptoms due to the very demanding [24], uncertain [44], and stressful situation. Nevertheless, it is reasonable to assume that such psychological outcomes are also due to more specific contextual conditions (first of all, as argued above, having worked on the frontline or not), as well as to individual differences in personality traits, ability to tolerate uncertainty and to cope with it. Thus, the main aim of the present paper was to investigate, in a sample of 682 Italian doctors and nurses, whether specific job conditions, i.e., having worked in a dedicated COVID ward (280) or not (402) during the first wave of the pandemic in Italy, would have determined differences in the associations (i.e., in the structure of relations) between personality traits, intolerance to uncertainty, coping strategies, and perceived stress. 

In line with our expectations, the analysis (LMMs) did not reveal significant differences between the two groups of Italian HCWs with regard to personality traits, intolerance of uncertainty, and coping strategies. In other words, the two groups of HCWs appear to be homogeneous not only in terms of dispositional traits, but also in terms of coping strategies adopted to face the situation, at least in the first phases of the pandemic. Conscientiousness and high levels of prospective intolerance of uncertainty seem to characterize our sample of HCWs. Both the ability to be organized, responsible, and productive (typical of the conscientious personality) and to not be paralyzed by uncertainty—but on the contrary to be engaged in information seeking, which is a typical trait of the prospective intolerance of uncertainty—seem to be focal in HCWs’ work. Furthermore, resorting to coping strategies mainly focused on emotions (acceptance, seeking emotional support, humor, positive reframing, religion) seems understandable in a situation such as a pandemic, which, especially in its early stages, was characterized by high levels of uncertainty and unpredictability, and which confronted health workers every day with suffering and death. 

Nonetheless, the analysis (LM) revealed higher levels of perceived stress among the frontline HCWs rather than in the non-frontline. This finding is consistent with the results of previous studies on both past epidemics [99] and the COVID-19 pandemic [17]. Specifically, during the outbreak in 2020, both Italian [100,101] and Chinese [21] frontline HCWs reported high levels of perceived stress and were more exposed to psychological burden than second line HCWs in terms of anxiety, depression, insomnia, and distress [25]. Moreover, this data seem to be understandable in the light of the increased risks faced by the frontline HCWs. 

Multi-group path-analysis based on the lavaan R-software package was used and the results mainly confirmed our hypotheses, revealing for the two groups of Italian HCWs different models of associations among the variables taken into account (see Figure 5A,B). Specifically, the analysis revealed more complex associations in the non-frontline HCWs’ group. This could be due to the greater heterogeneity of this group of healthcare professionals who, unlike the group who worked in dedicated COVID wards, continued to work in different types of wards (which are characterized per se by an intrinsic diversity).

Specifically, significant differences were found between frontline and non-frontline Italian HCWs concerning the associations between: personality traits and intolerance of uncertainty; intolerance of uncertainty and coping strategies.

As for the associations between personality traits and intolerance of uncertainty, among the non-frontline HCWs, high levels of conscientiousness are negatively related to prospective intolerance of uncertainty. In other words, the more they are conscientious (i.e., able from an organizational, productive, and responsible point of view), the less they show need and desire for predictability and active engagement in increasing their certainty. High levels of agreeableness and open-mindedness are, instead, in the same group, negatively related with inhibitory intolerance of uncertainty. In other words, the more individuals are agreeable (i.e., confident and compassionate) and open-minded (i.e., curious and open to the unexpected events), the less they seem to be paralyzed by inhibitory uncertainty. Vice versa, among the frontline HCWs, personality traits seem to have had a poor influence on intolerance of uncertainty, except for neuroticism, that seems to act similarly in both groups of HCWs by increasing both the prospective and inhibitory intolerance of uncertainty.

As for the associations between intolerance of uncertainty and coping strategies, no significant relationship has been identified among the frontline HCWs. It is as if knowing with certainty dealing with infected people reduces the effect of the intolerance of uncertainty on coping strategies. On the contrary, among the non-frontline HCWs, prospective intolerance of uncertainty is positively linked with problem and emotion-focused coping strategies. In other words, HCWs with high levels of prospective intolerance of uncertainty resort to problem and emotion-focused strategies in facing the situation, i.e., acting in order to reduce uncertainty. In this sense, prospective uncertainty seems to be predictive of greater use of functional coping strategies among health professionals. Vice versa, inhibitory intolerance of uncertainty is negatively related to problem and emotion-focused coping strategies (in line with its paralyzing traits) and positively related with dysfunctional coping strategies (in line with its avoidance characteristics). In other words, higher levels of inhibitory intolerance of uncertainty seem to be predictive of greater recourse to dysfunctional coping strategies. In this sense, inhibitory intolerance of uncertainty seems to have a more negative impact on HCWs’ ability to cope with stressful situations rather than the prospective one.

Interesting similarities were found instead between the two groups of HCWs regarding the role of negative emotionality (neuroticism) in affecting intolerance of uncertainty, and concerning the association between coping strategies and perceived stress.

The finding according to which negative emotionality (neuroticism) affects intolerance of uncertainty is in line with the results of much research, mainly concerning general population samples [49,59,102,103], also during the pandemic [104], according to which “poor emotional regulation skills contribute to intolerance to uncertainty” [105] (p. 4). Irrespective of having worked on the frontline or not, Italian HCWs with high levels of neuroticism also showed high levels of intolerance of uncertainty (both prospective and inhibitory). In other words, neuroticism seems to be predictive of high levels of intolerance of uncertainty also among healthcare professionals irrespective of their being frontline or not. 

As for the association between coping strategies and perceived stress, in both groups, resorting to emotion-focused coping strategies (acceptance, seeking emotional support, humor, positive reframing, religion) was negatively related to perceived stress, whereas the dysfunctional one was positively linked to stress. The first association we identified, according to which emotion-focused coping strategies are linked to lower level of perceived stress, thus functioning as a protective factor against negative psychological outcomes, is consistent with the results of many other studies on the general population during COVID-19 [106], as well as on HCWs before COVID-19 [107,108,109] and during it [110,111]. Similarly, also the second association we identified, according to which, on the contrary, dysfunctional coping strategies (behavioral disengagement, denial, self-blame, self-distraction, substance use) are linked to higher levels of perceived stress, is consistent with the results of many other studies [79,100,110], as well as in line with our expectations. Furthermore, consistently with the results of other works on Italian HCWs employed in facing COVID-19 during the first months of its spread [101,112,113,114], we did not find positive associations between problem-focused coping strategies and stress reduction in both groups of HCWs. Analogously to the results of these studies [101,112,113,114], our findings revealed that, problem-focused coping strategies were not effective in the reduction of HCWs’ perceived stress during the first wave of the pandemic (that was the period the respondents of our questionnaire referred to) probably due to the “lack of scientific knowledge about the therapeutic and treatment procedures effective for COVID-19” [114] (p. 3). In other words, the insufficient knowledge and the wide-spread uncertainty about the effective procedures to apply in order to prevent the spread of the virus and to treat infected people, seem to have made it difficult for health professionals to resort to problem-focused coping as strategies for stress reduction.

There are some limitations to the current research that should be considered. The first one concerns methodology: the research used self-report measures (although exclusively validated scales has been utilized, they can lead to potential bias related to social desirability), involved a non-probabilistic sample and it was a cross-sectional study. Moreover, although information has been collected on doctors and nurses, the level of severity of the patients with whom the participants had been in contact has not been specified. Furthermore, other significant variables such as years of service (seniority), age, medical specializations, gender, etc., have not been taken into account. Finally, doctors and nurses were not randomly assigned to workplaces. In addition, we did not evaluate the role played by reasoning processes, i.e., by cognitive strategies used by HCWs to reach decisions. While non-frontline HCWs (who generally did not have to deal with virus-related emergencies) probably have had more time to process information, to evaluate possible alternatives, and to make decisions, frontline HCWs more likely have had less time to think, to consider alternatives, and to assume decisions. This may have led the frontline HCWs to resort more frequently to shortcuts in thinking (heuristics) [115,116], which may have had some influence on the associations between the variables we examined, perhaps even inhibiting or reducing the strength of dispositional traits and the associations between them. Nonetheless, since we did not take into consideration reasoning processes, their possible impact remains a supposition, which deserves to be explored in future research.

Future studies might also take into account how the socio-demographic and work-related variables impact stress, as well as other psychological outcomes, among HCWs. Furthermore, it would also be interesting to investigate HCWs’ point of views, i.e., to analyze not only their responses to the items of validated scales (using quantitative methods), but also to analyze (using qualitative methods), their open-ended responses and/or interviews. Regarding this last point, our research team is qualitatively analyzing a sample of responses given to the last open-ended question of our questionnaire, which aimed to know if and how the experience of working during the pandemic had an emotional impact on HCWs (see Section 2.1). It would have been interesting also to repeat the survey after the second wave of the virus outbreak to test if new knowledge concerning the virus spread and its cure had influenced the use of coping strategies and the levels of perceived stress among Italian HCWs, and if so, how.

Despite the limitations, the results of the current study might be useful for planning and adopting preventing approaches to reduce HCWs’ stress burden during a health emergency. The inability to tolerate uncertainty or the use of dysfunctional coping strategies, in fact, not only lead to negative outcomes for HCWs, but may also have an impact on patients and healthcare systems. The planning of training courses aimed to provide HCWs with skillsets they can use to cope with uncertain and stressful situations (such as that related to the COVID-19 pandemic) might be effective not only in reducing and controlling perceived stress (thus improving their mental wellbeing), but also in improving HCWs’ effectiveness, and, thus have positive impacts on patients’ health and on reducing costs for healthcare systems. Effective interventions should be designed to fit the specific traits of HCWs at the forefront. Health professionals who are better equipped (in psychological terms) to cope with uncertain and stressful situations would undoubtedly lead to improve the quality of care.

## Figures and Tables

**Figure 1 healthcare-09-01086-f001:**
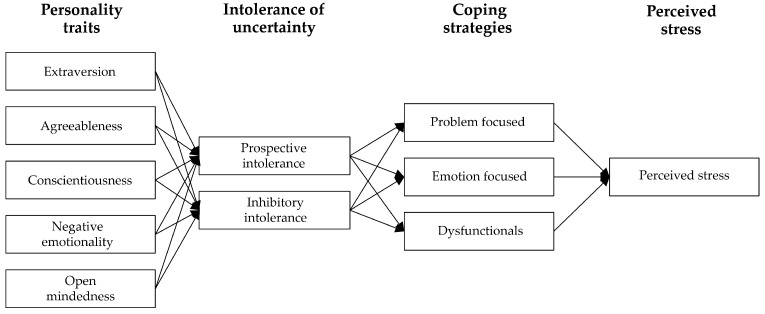
Path-diagram of the general model of structure of relations among variables.

**Figure 2 healthcare-09-01086-f002:**
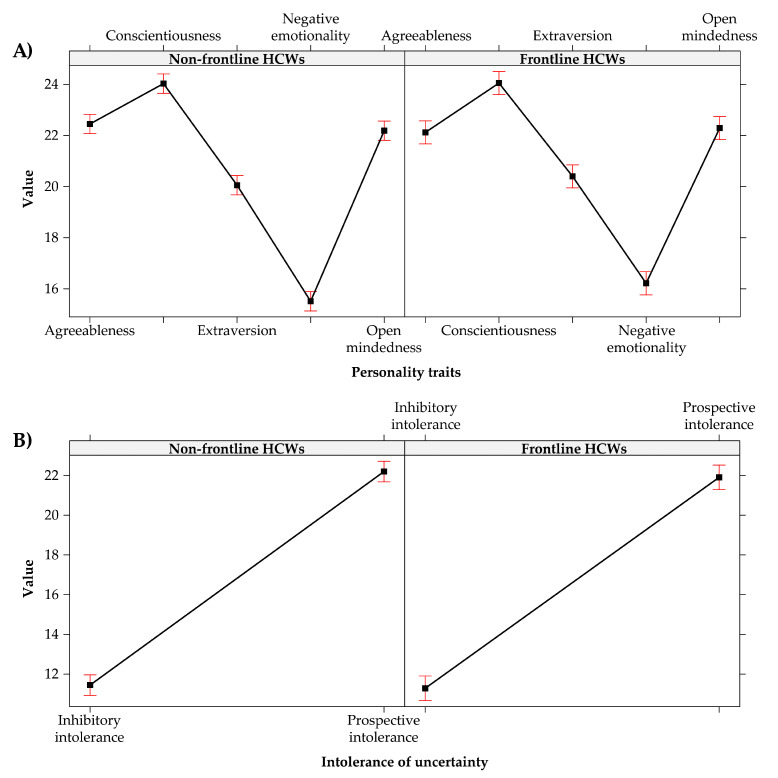
Effect plot of: (**A**) personality traits; (**B**) intolerance of uncertainly. The bars represent the 95% CI.

**Figure 3 healthcare-09-01086-f003:**
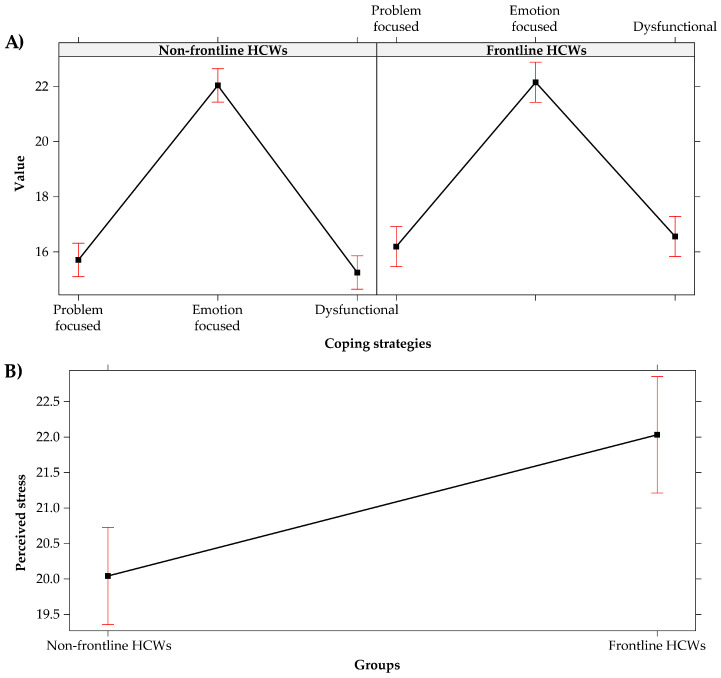
Effect plot of: (**A**) coping strategies; (**B**) perceived stress of groups. The bars represent the 95% CI.

**Figure 4 healthcare-09-01086-f004:**
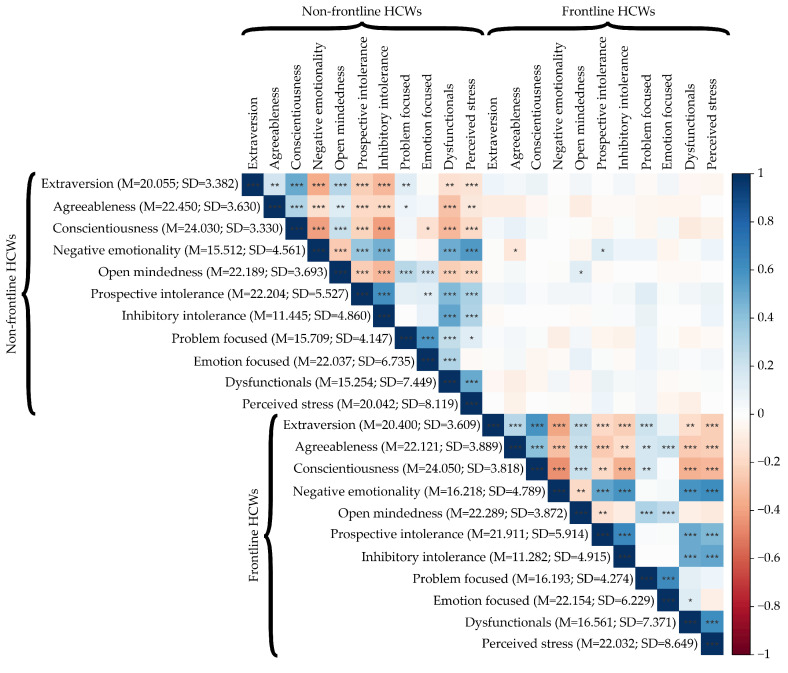
Pearson’s Correlations and descriptive statistics of the experimental variables (* *p* < 0.05, ** *p* < 0.01, *** *p* < 0.001).

**Figure 5 healthcare-09-01086-f005:**
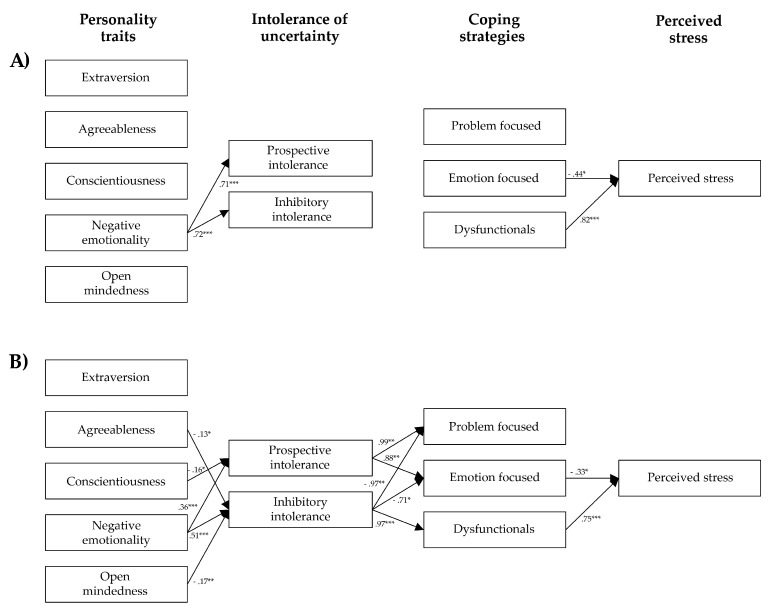
Structure of relations in the: (**A**) Frontline group of Italian HCWs; (**B**) non-frontline group of Italian HCWs. Signif. codes: 0 “***” 0.001, “**” 0.01, “*” 0.05, “.” 0.1.

**Table 1 healthcare-09-01086-t001:** Descriptive statistics of the sample characteristics.

Variables	n (%)
**Total**	682 (100.00%)
**Socio-demographic characteristics**	
*Gender*	
Female	530 (77.71%)
Male	152 (22.29%)
*Age*	
18–30	128 (18.77%)
31–40	131 (19.21%)
41–50	182 (26.69%)
51–60	193 (28.30%)
>60	48 (7.04%)
*Marital status*	
Married	307 (45.01%)
Unmarried	188 (27.57%)
Domestic partner	107 (15.69%)
Divorced/separated	67 (9.82%)
Widower/widow	13 (1.91%)
*Children*	
Yes	395 (57.92%)
No	287 (42.08%)
*Religion*	
Believer occasionally practitioner	262 (38.42%)
Believer non-practitioner	162 (23.75%)
Non-Believer	113 (16.57%)
Believer practitioner	108 (15.84%)
Prefer not to answer	37 (5.43%)
**Job characteristics**	
*Place of work*	
North Italy	479 (70.23%)
Centre Italy	128 (18.77%)
South Italy	75 (11.00 %)
*Job position*	
Nurse	518 (75.95%)
Physician	164 (24.05%)
*Job area*	
Medical specialties	353 (51.76%)
Diagnostic and therapeutic specialties	144 (21.11%)
Surgical specialties	106 (15.54%)
Primary care nurse. serv.	79 (11.58%)
*Seniority*	
More than 20 years	325 (47.65%)
Less than 5 years	150 (21.99%)
10–20 years	121 (17.74%)
5–10 years	86 (12.61%)
**Job exposure to COVID-19**	
*Wards*	
Worked in COVID-19-dedicated wards	280 (41.06%)
Worked in other wards	402 (58.94%)
*Swabs for COVID-19*	
Done	392 (57.48%)
Not done	290 (42.52%)
*COVID-19 contracted*	
No	534 (78.30%)
Perhaps	91 (13.34%)
Yes	57 (8.36%)

**Table 2 healthcare-09-01086-t002:** Differences between frontline and non-frontline HCWs.

Model	Df	Chi-sq	Chi-sq Difference	Df Difference	*p*-Value	CFI Difference	TLI Difference	RMSEA Difference	SRMR Difference
Unconstrained	44	131.810	-	-	-	-	-	-	-
Constrained	70	177.990	46.180	26	0.008 **	0.009	−0.019	0.009	−0.008

Signif. codes: “**” 0.01.

## Data Availability

All relevant data presented in the study are included in the article. The datasets analyzed are available from the corresponding author on reasonable request.

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
