# Peer review of "Associations between Personality Traits, Intolerance of Uncertainty, Coping Strategies, and Stress in Italian Frontline and Non-Frontline HCWs during the COVID-19 Pandemic—A Multi-Group Path-Analysis"

_healthcare, 2021, doi:10.3390/healthcare9081086_

Round 1

Reviewer 1 Report

This paper presents information on individual traits, coping, and stress in healthcare workers during the Covid-10 pandemic. One general comment would be on the use of the past tense for this paper. I would think that the present tense would be more appropriate given that the pandemic is ongoing – however I defer to the editor. Another general comment is that there were some very long sentences and some grammatical issues, but nothing too significant. Overall, this was an interesting paper and I think the authors have done well.

Abstract

  • It would be helpful if the implications of the findings could be described, rather than simply stating that there are implications.

Introduction

  • Grammar in the first paragraph
  • It would be good to see the link made between coping and personal traits with stress prior to the description of the model. i.e., why would these factors impact stress? A little more of a theoretical framework may be useful in this section.
  • Some of the information provided in the intro may be better suited to the methods i.e., what specifically was done in this study. It would be better to stay focused on the hypotheses here.

Methods

  • I am not sure what the authors mean by ‘related test’ in the following list: “exposure to COVID-19, related test, and contraction of COVID-19.” (ln222)
  • It is interesting that personality was a key part of the method, but it seems from the introduction that intolerance to uncertainty was the main trait being examined. Perhaps this could be clarified a little more in the introduction.

Results

  • This may just be a viewer issue, but the font of the table on p11 and 12 was very small and hard to read!
  • Section 3.3 – suggest including the path analysis figure here. This section was a little hard to follow – could more explanation of the analyses and outcomes be provided?

Discussion

  • I suggest starting the discussion with an overview of the findings rather than going over the background again. It would be good to start with the key findings (i.e., the comparison between frontline and non frontline workers).
  • The rationale surrounding heuristics could be more clearly explained.
  • Unclear why dot points are used for some paragraphs
  • The authors state “The results of the current study have significant implications for developing measures against negative psychological repercussions for HCWs”. However, these implications have not been explained in detail.

Author Response

Dear reviewer,

thank you for your valuable revisions. We replied point by point to your suggestions.  Tou can read our replies in capital letter and in red colour.

Comments and Suggestions for Authors

This paper presents information on individual traits, coping, and stress in healthcare workers during the Covid-10 pandemic. One general comment would be on the use of the past tense for this paper. DONE. WE REPLACED THE PAST WITH THE PRESENT TENSE IN THE INTRODUCTION. I would think that the present tense would be more appropriate given that the pandemic is ongoing – however I defer to the editor. Another general comment is that there were some very long sentences and some grammatical issues, but nothing too significant. Overall, this was an interesting paper and I think the authors have done well.

Abstract

  • It would be helpful if the implications of the findings could be described, rather than simply stating that there are implications  WE MODIFIED THE LAST PART OF THE ABSTRACT. WE HOPE NOW IT IS CLEARER
  • Introduction
  • Grammar in the first paragraph  DONE. WE REVISED SENTENCES
  • It would be good to see the link made between coping and personal traits with stress prior to the description of the model. i.e., why would these factors impact stress? A little more of a theoretical framework may be useful in this section. WE ADDED SOME THEORETICAL FRAMEWORK ON THE TOPIC
  • Some of the information provided in the intro may be better suited to the methods i.e., what specifically was done in this study. It would be better to stay focused on the hypotheses here. WE MOVED THE DESCRIPTION OF THE PROCEDURES AND FIGURE 1 IN THE METHOD SECTION.

Methods

  • I am not sure what the authors mean by ‘related test’ in the following list: “exposure to COVID-19, related test, and contraction of COVID-19.” (ln222) WE REFER TO SWABS FOR COVID-19. WE MODIFIED IN THE TEXT AND IN THE TABLE.
  • It is interesting that personality was a key part of the method, but it seems from the introduction that intolerance to uncertainty was the main trait being examined. Perhaps this could be clarified a little more in the introduction. DONE

Results

  • This may just be a viewer issue, but the font of the table on p11 and 12 was very small and hard to read! WE REPLACED THE TABLE WITH A FIGURE (4).
  • Section 3.3 – suggest including the path analysis figure here. WE MOVED FIGURE 1 IN A NEW SECTION (2.3) DEVODET TO THE PROCEDURES, FOLLOWING YOUR PREVIOUS SUGGESTION “Some of the information provided in the intro may be better suited to the methods i.e., what specifically was done in this study. This section was a little hard to follow – could more explanation of the analyses and outcomes be provided? WE HOPE THAT THE NEW INFORMATION AND FIGURES HAVE MADE THE SENSE OF THE RESULTS CLEARER

Discussion

  • I suggest starting the discussion with an overview of the findings rather than going over the background again. It would be good to start with the key findings (i.e., the comparison between frontline and non frontline workers). WE HAVE LEFT THE INITIAL PART OF THE DISCUSSION BECAUSE SOMETIMES READERS FOCUS ON THE ABSTRACT, INTRODUCTION, AND DISCUSSION AND IT MIGHT BE EASIER FOR THEM TO GET AN OVERVIEW OF THE CONTENT. HOWEVER, IF YOU THINK IT IS BETTER TO REMOVE THE FIRST PART, WE AGREE TO REMOVE THE FIRST TWO PARAGRAPHS (I.E. FROM "COVID-19"... TO "AND PERCEIVED STRESS").
  • The rationale surrounding heuristics could be more clearly explained WE SYNTHETIZED AND MOVED THIS PART IN PARAGRAPHS DEVOTED TO LIMITATIONS AND FUTURE AVENUES.
  • Unclear why dot points are used for some paragraphs WE REMOVED THEM.
  • The authors state “The results of the current study have significant implications for developing measures against negative psychological repercussions for HCWs”. However, these implications have not been explained in detail. WE MODIFIED also ACCORDING TO WHAT WE WROTE IN THE ABSTRACT

Reviewer 2 Report

Comments

  1. The empirical context of the study should be motivated in the introduction.
  2. The measures that are used in the empirical analysis are self-reported. This may limit the conclusions from the data.
  3. The paper should provide information about non-response.
  4. Employees are not randomly assigned into workplaces. Failure to account for sorting of employees may bias any estimated effects (https://doi.org/10.1016/j.jebo.2012.09.005). This issue should be noted in the revised version.
  5. The revised version should put more emphasis on the quantitative size of the estimated effects.
  6. The paper does not consider the heterogeneity in the estimated effects. The relationships between the variables of interest can differ significantly, e.g., by age. The relatively small size limits the analysis by subgroups.
  7. The concluding section of the paper should state useful avenues for future studies.

Author Response

Dear reviewer,

thank you for your valuable revisions.

We tried to modify the paper according to your suggestions. 

We replied point by point to your suggestion by using capital letters and red color.

Comments and Suggestions for Authors

Comments

  1. The empirical context of the study should be motivated in the introduction. DONE. WE ADDED SOME SENTENCES IN ORDER TO BETTER MOTIVATE
  2. The measures that are used in the empirical analysis are self-reported. This may limit the conclusions from the data. DONE. WE INCLUDED THIS POINT IN THE METHODOLOGICAL LIMITATIONS at the end of the paper.
  3. The paper should provide information about non-response. DONE. 
  4. Employees are not randomly assigned into workplaces. Failure to account for sorting of employees may bias any estimated effects (https://doi.org/10.1016/j.jebo.2012.09.005). This issue should be noted in the revised version. DONE.
  5. The revised version should put more emphasis on the quantitative size of the estimated effects. DONE.
  6. The paper does not consider the heterogeneity in the estimated effects. The relationships between the variables of interest can differ significantly, e.g., by age. The relatively small size limits the analysis by subgroups. DONE
  7. The concluding section of the paper should state useful avenues for future studies. DONE.